# The Structure of Relation Decoding Linear Operators in Large Language Models

**Miranda Anna Christ**[*1,2], **Adrián Csiszárik**[*2,3], **Gergely Becsó**[2,3], **Dániel Varga**[2]

[1]Fazekas Mihály High School, Budapest, Hungary
[2]HUN-REN Alfréd Rényi Insititute of Mathematics, Budapest, Hungary
[3]Eötvös Loránd University, Budapest, Hungary
chrm@berkeley.edu, {csadrian, begeri, daniel}@renyi.hu

## Abstract

This paper investigates the structure of linear operators introduced in Hernandez et al. [2023] that decode specific relational facts in transformer language models. We extend their single-relation findings to a collection of relations and systematically chart their organization. We show that such collections of relation decoders can be highly compressed by simple order-3 tensor networks without significant loss in decoding accuracy. To explain this surprising redundancy, we develop a cross-evaluation protocol, in which we apply each linear decoder operator to the subjects of every other relation. Our results reveal that these linear maps do not encode distinct relations, but extract recurring, coarse-grained semantic properties (e.g., *country of capital city* and *country of food* are both in the *country-of-X* property). This property-centric structure clarifies both the operators' compressibility and highlights why they generalize only to new relations that are semantically close. Our findings thus interpret linear relational decoding in transformer language models as primarily property-based, rather than relation-specific.[1]

## 1 Introduction

From ancient philosophy to modern cognitive science, knowledge has been framed not merely as isolated entities but as interconnected networks shaped by context and structure. Aristotle's categorical distinctions, Wittgenstein's concept of family resemblances [Wittgenstein, 1953], and contemporary cognitive theories of prototypes [Rosch, 1973] all emphasize that understanding emerges from recognizing shared properties across diverse contexts. We do not simply store isolated facts, instead, we perceive who does what to whom, what belongs where, and how different ideas interrelate through common attributes. Investigating the underlying structure of how relational knowledge is encoded thus becomes crucial, as it forms the foundation for complex cognitive functions such as generalization, abstraction, and analogical reasoning—capabilities essential for both human cognition and machine learning models.

In this paper, we investigate how transformer language models encode relational knowledge—such as *Michael Jordan plays basketball*—by mapping subjects directly to objects through linguistic predicates. Our investigation builds on the recent work of Hernandez et al. [2023], who show that for a given relation, the transformation from the embedding of a subject to the tokens of the object can be effectively approximated by a single linear operator, called a Linear Relational Embedding (LRE) matrix. Here, we examine relations through the lens of their corresponding linear decoder

---

[*]Equal contributions.

[1]Code and data are available at the project website: https://bit.ly/structure-of-relations.

39th Conference on Neural Information Processing Systems (NeurIPS 2025).

operators, and perhaps more crucially, rather than analyzing them one-by-one, we consider them *collectively*. We develop tools to uncover their structure and the underlying regularities with the goal of understanding their collective organization.

Beyond interpretability, we are also motivated by the possibility that these decoder operators admit a more compact, shared representation. Such compression could highlight the core semantic properties shared across different relations, reducing redundancy and perhaps enabling the model to generalize more effectively, analogously to how humans draw on abstract patterns to reason across different contexts. Our experiments with tensor network models explore this hypothesis by seeking compact representations that preserve decoding capacity while exploiting the possible latent structure of relational knowledge.

We summarize our contributions as follows:

- We propose a novel semantic closeness notion to explore the underlying structure of relations. We reveal that relation decoding functions have a property-based rather than a fine-grained relation-specific organization.

- We propose order-3 tensor networks to compress an entire collection of linear decoder functions into a single, compact model. We demonstrate and analyze the effectiveness of this approach.

- We investigate the generalization properties of such tensor network models. We find low level generalization capabilities on generic data, and also provide examples where they excel.

## 2   Background and Notation

### 2.1   Relations in Transformer Language Models

One common approach to formalize factual knowledge in language models is through relational triplets $(S, R, O)$, where a subject $S$ is linked to an object $O$ via a relation $R$ [Miller, 1995, Berners-Lee et al., 2001, Bollacker et al., 2008, Lenat, 1995, Richens, 1956, Minsky, 1974]. Here, $S \in \mathcal{S}$, $O \in \mathcal{O}$, and $R \in \mathcal{R}$, with $\mathcal{S}$, $\mathcal{O}$, and $\mathcal{R}$ denoting the sets of all possible subjects, objects, and relations, respectively. For instance, the sentence "Paris is the capital city of France" corresponds to the triplet ("Paris", "capital city of", "France").

Recent work by Hernandez et al. [2023] demonstrated that for each relation $R$, the transformation from subject embeddings to object embeddings in transformer-based language models can be effectively modeled by a surprisingly simple decoder: an affine transformation $f_R : \mathcal{S} \to \mathcal{O}$, $f_R(S) = W_R \boldsymbol{v}_S + \boldsymbol{b}_R$, where $W_R \in \mathbb{R}^{d \times d}$ and $\boldsymbol{b}_R \in \mathbb{R}^d$, and $d \in \mathbb{N}$ is the embedding dimension of the transformer, and $\boldsymbol{v}_S$ is the embedding of the subject. The effectiveness of such affine approximations suggest a surprisingly simple internal organization for relational knowledge within transformer layers.

### 2.2   Technical Details

Throughout our experiments we study LLMs based on the transformer architecture [Vaswani et al., 2017]. The model maps input tokens into a $d$-dimensional space via an embedding layer $L_{\text{emb}}$. This is followed by a sequence of $H \in \mathbb{N}$ transformer blocks, denoted as $L_h$ ($h \in [H]$), and concludes with a final transformer head $L_{\text{head}}$. The entire transformer network is thus the function $t = L_{\text{emb}} \circ L_1 \circ L_2 \circ \cdots \circ L_{k-1} \circ L_k \circ L_{\text{head}}$. Approximating an affine relation decoder thus equals to modeling the function $g = L_l \circ L_{l+1} \circ \cdots \circ L_{k-1} \circ L_k$ (i.e., the function of the transformer from the $l$th layer until $L_{\text{head}}$) with an affine transformation, where k is the total number of blocks before the $L_{\text{head}}$ component.

One could find many ways to infer an affine approximation of the function $g$. In this paper, we infer it with a **parameterized modeling function**, instead of directly approximating it using the Jacobian of $g$ (as shown in Hernandez et al. [2023]).

Once we have a linear relation embedding (LRE) matrix $W_R$ and a bias $\boldsymbol{b}_R$, we apply the affine function $f_R(\boldsymbol{v}_S) = W_R \boldsymbol{v}_S + \boldsymbol{b}_R$ to the subject embedding $\boldsymbol{v}_S$. Then we pass the resulting vector to the transformer's output head to generate the next token. The performance of the relation decoding function is evaluated using the notion of *faithfulness*, which essentially measures prediction accuracy.

**Definition 1 (Faithfulness)** *Given a subject token embedding $v_S \in \mathbb{R}^d$ and an inferred linear relation encoder matrix $W_R \in \mathbb{R}^{d \times d}$ and bias $b_R \in \mathbb{R}^d$ for a relation $R$, we define the* faithfulness *of the affine decoder $f_R(v_S) = W_R v_S + b_R$ as the top-1 accuracy of the predicted object token. Specifically, we compute $\hat{v}_O = L_{head}(f_R(v_S))$, where $L_{head}$ is the transformer's output head, and compare the resulting token prediction against the ground truth object token. Faithfulness is then the proportion of correct predictions across a set of subject-relation-object triplets.*

## 2.3 Models and Datasets

We used three transformer-based language models in our experiments: GPT-J [Wang and Komatsuzaki, 2021], Llama 3.1 8B [Dubey and et al., 2024], and GPT-NeoX-20B [Black et al., 2022b]. Unless stated otherwise, the results presented in the main text correspond to GPT-J, with additional analyses and details for the other models provided in Appendix C.

We use three datasets: **1) The Dataset of Hernandez et al. [2023]**: it consists of 47 distinct, mostly orthogonal relations (i.e., *fruit inside color* and *adjective antonym*). **2) Extended Dataset**: our extended version of the dataset of Hernandez et al. [2023] that introduces several new relations, allowing a better understanding of the relational structure. **3) Mathematical Dataset**: a novel relational dataset containing mathematical operations (i.e., *number plus 6* and *number times 9*) providing a more controlled, and in a sense a denser relational structure. For further information, we refer to Appendix D.

## 3 Training Tensor Networks to Represent a Collection of Relations

Matrices representing relations live in $\mathbb{R}^{d \times d}$, where $d$ is the dimension of the embedding space. In the case of GPT-J, $d = 4096$, meaning even a single matrix has more than 16 million parameters. With a collection of 100 relations that amounts to approximately 1.6 billion parameters. In this section, we explore the following question:

> *Is it possible to represent a collection of relations in a compact form, and can this representation rely on the striking simplicity of linearity?*

**From the naïve representation to tensor networks**   A straightforward way to represent multiple relation matrices is to stack them to form an order-3 tensor. Specifically, a collection of $n$ matrices each of size $\mathbb{R}^{d \times d}$, can naturally be regarded as a single 3-tensor of dimensions $\mathbb{R}^{d \times d \times n}$. A primary approach to 'compress' matrices in a linear way is via low-rank approximation. A more general, tensor-analogous approach would involve decomposing this order-3 tensor to produce a representation that retains the information content and the functionality of the original tensor, while using significantly fewer parameters. The notion of *tensor networks* offers a principled way and serves as a remarkably powerful tool for this endeavor: any tensor network having three free legs of dimension $d$ can be viewed as a representation of an order-3 tensor. (Regarding tensor networks, we restrict ourselves to introducing only the minimal terminology required for our models in Appendix F. For a more detailed introduction we refer to Ahle [2024].) Furthermore, inner dimension constraints serve as bottlenecks—analogously to rank constraints—significantly reducing the parameter count. The number of possible internal structures of '3-legged' tensor networks is vast (in fact, theoretically infinite). While this richness of possibilities opens up a plethora of opportunities worth for exploration, for the sake of a compact exposition, here we explore only two of these.

### 3.1 Tensor Network Architectures

A *tensor network* is a collection of tensors $\mathcal{T} = \{T^1, T^2, \ldots, T^k\}$ with a pairing of the legs of the tensors, where each pair connects legs of equal dimensions. Unpaired legs are called 'free legs' and their number is called the order of the network. The 'free legs' of a tensor network are often simply referred to as 'legs'. We can draw these networks as tensor diagrams: dots are tensors, lines are their legs, and joined legs link tensors like edges in a graph. Each leg has a label. One labeled picture therefore captures both the layout and the workings of the whole tensor network. In this paper, we implement two tensor network classes. See Fig. 1b for the diagrams of these tensor networks.

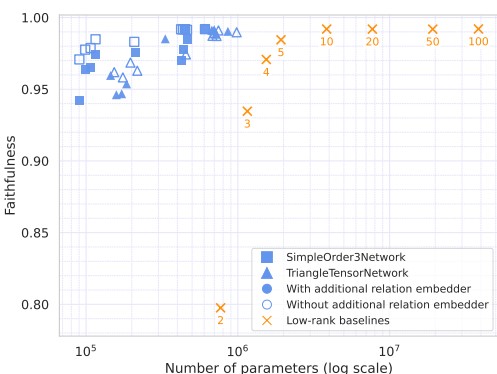

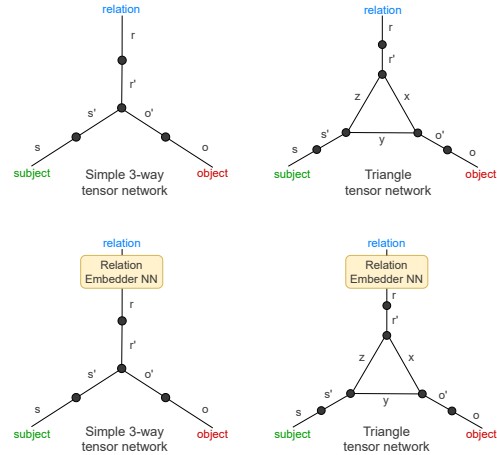

(a) Average faithfulness of a collection of compressed linear operators in the function of paremeter count. Each point corresponds to a trained tensor network. Filled markers represent models with additional relation embedders. Different marker figures represent tensor networks with different inner structure (see Figure 1b on the right). We employ a low-rank baseline — element-wise low-rank approximation of the decoders.

(b) Schematic illustrations of the tensor networks used in this paper. Left: a basic order-3 tensor network. Right: a triangular configuration of a tensor network. Both architectures can optionally incorporate a separate relation embedding network.

Figure 1: Compressing relation decoding linear operators with order-3 tensor networks.

**SimpleOrder3Network** The network consists of an order-3 tensor $T^0 \in \mathbb{R}^{d_{s'} \times d_{r'} \times d_{o'}}$ at the core. To connect it with the embeddings of the transformer embeddings, we use three order-2 tensors (i.e., matrices) $P^1_{s,s'} \in \mathbb{R}^{d \times d_{s'}}, P^2_{r,r'} \in \mathbb{R}^{d \times d_{r'}}, P^3_{o,o'} \in \mathbb{R}^{d \times d_{o'}}$ to project the inputs into the appropriate space. We call $d_{s'}$, $d_{r'}$ and $d_{o'}$ the inner dimensions of the network. The relation-to-matrix mapping implemented by this network can be written out as

$$T^R_{s,o} = T_{s,r,o}\boldsymbol{v}_r = \sum_{r,s',r',o'} \boldsymbol{v}_r P^2_{r,r'} T^0_{s',r',o'} P^1_{s,s'} P^2_{o,o'},$$

where $T^R_{s,o}$ denotes the relation decoder approximation obtained from the tensor network $T$ and relation embedding $\boldsymbol{v}_r$ for relation $R$.

**TriangleTensorNetwork** The network consists of three inner order-3 tensors $T^1_{s',y,z}, T^2_{x,r',z}, T^3_{x,y,o'}$ in $\mathbb{R}^{d_{s'} \times d_y \times d_z}, \mathbb{R}^{d_x \times d_{r'} \times d_z}, \mathbb{R}^{d_x \times d_y \times d_{o'}}$ respectively, and three 2-tensors (matrices) $P^1_{s,s'} \in \mathbb{R}^{d \times d_{s'}}, P^2_{r,r'} \in \mathbb{R}^{d \times d_{r'}}, P^3_{o,o'} \in \mathbb{R}^{d \times d_{o'}}$ projecting the transformer embeddings into the appropriate space. Similarly as above, $d_{s'}$, $d_{r'}$ and $d_{o'}$ are the inner dimensions of the network, serving as a bottleneck. The vector-to-matrix mapping implemented by this network can be written out as:

$$T^R_{s,o} = T_{s,r,o}\boldsymbol{v}_r = \sum_{r,s',r',o',x,y,z} \boldsymbol{v}_r P^2_{r,r'} T^1_{s',y,z} T^2_{x,r',z} T^3_{x,y,o'} P^1_{s,s'} P^2_{o,o'}.$$

**Parameter counts** The total number of parameters depends on the embedding dimension $d$ of the language model and the inner dimensions $d_{s'}$, $d_{r'}$, and $d_{o'}$ of the core tensors. For the *Simple-Order3Network*, the parameter count is $N_{\text{Simple}} = (d \cdot d_{s'} + d \cdot d_{r'} + d \cdot d_{o'}) + (d_{s'} \cdot d_{r'} \cdot d_{o'})$. The first term corresponds to the three projection matrices, which dominate the total parameter count, while the second term represents the compact order-3 core tensor. For the *TriangleTensorNetwork*, the total parameters are $N_{\text{Triangle}} = (d \cdot d_{s'} + d \cdot d_{r'} + d \cdot d_{o'}) + 3 \cdot (d_{s'} \cdot d_{r'} \cdot d_{o'})$, reflecting three interconnected order-3 tensors in the core.

**From matrices to affine maps** The networks above produce matrices—linear maps with no bias—but our task need to produce affine maps. We can embed the bias term directly in the network by enlarging the appropriate bond dimension with 1. At inference time, append a constant 1 to the subject vectors, during training the extra column of weights learns the bias jointly with the rest of the matrix. This simple tweak converts the linear output into an affine map without altering the overall architecture.

**Additional embedder network on the relation leg**   Optionally, both architectures can be extended with a *Additional Relation Embedder* to loosen the rather strong assumption of linearity on the relation embeddings before entering the tensor network. We experimented with an embedder consisting of three dense feed-forward layers with ReLU [Agarap, 2018] activations.

## 3.2   Training Tensor Networks

A tensor network even with a fixed structure can still be obtained and utilized in various ways. Tensor parameters may be determined through factorization or optimization algorithms. Once constructed, the resulting tensor network can be used by evaluating contractions on its free legs.

**End-to-end training of tensor networks**   We fix the parameters of the LLM and train only the tensor network. We train our tensor network using the task loss, i.e., the cross-entropy loss for the predicted object token and optimize with SGD. Specifically, **1)** we take $v_R$, the embedding of the relation, **2)** perform a contraction with the tensor network at the relation leg, **3)** read out the matrix representing the LRE, **4)** apply it to the subject embedding $v_S$, then **5)** pass it to the $L_{\text{head}}$ to get the predicted token distribution, **6)** use the cross-entropy loss to match the expected token as a loss function. Formally, our loss function $\mathcal{L}_{\mathcal{R}}$ is:

$$\mathcal{L}_{\mathcal{R}}(T_{s,r,o}) = \sum_{R \in \mathcal{R}} \sum_{(S,O) \in R} CE(\mathbb{1}_O, L_{\text{head}}(T^R_{s,o}(\boldsymbol{v}_S))),$$

where $\mathbb{1}_O$ denotes the one-hot encoded first token of the object $O$, $CE$ is the cross-entropy loss and $v_S$ is a vector representation of subject $S$. Throughout our paper we always use the ground-truth object values for training.

## 3.3   Compression Experiments

In these experiments we evaluate the compression capabilities of both SimpleOrder3Network and TriangleTensorNetwork models. We train these models on the dataset of Hernandez et al. [2023] until convergence and measure the faithfulness of the resulting decoder functions. We performed a grid search for both models with $d_{r'} \in \{2, 4, 6, 8, 30, 100\}$, $d_{s'}, d_{o'} \in \{10, 50, 100, 300\}$; for the TriangleTensorNetwork we fixed $d_x, d_y, d_z \in \{50\}$. We also trained the tensor networks with and without an extra relation embedder. We discuss hyperparameters in Appendix 4.

**Baselines**   As a baseline, we train low-rank LRE matrices *individually*, producing one per relation. Each matrix is optimized independently using the same training objective as the tensor networks. This baseline corresponds to a low-rank representation of each relation matrix, without sharing parameters across relations or introducing any structured connectivity between them. In addition, as a canonical reference point, we note that a linear decoder baseline following the Jacobian-based procedure of Hernandez et al. [2023], which estimates each LRE from a few examples (we used 8 in our setup) achieves a mean faithfulness of 0.41 with approximately 788 million parameters.

**Relation decoding matrices are highly compressible**   Figure 1a summarizes the results by plotting the mean faithfulness against the parameter count of trained tensor network matrices. *We can clearly observe that the relation matrices are highly compressible.* Compared to the "vanilla representation" of stacking the 47 relation matrices together (with an overall parameter count of about 788 million) tensor networks even with less than one million parameters significantly outperform the baselines both in terms of parameter count and faithfulness. We can also observe that tensor network models without separate relation embedders consistently outperform those with them, showing that a linear structure is sufficient for efficient compression.

**Do LREs produced by tensor networks retain their sample-wise generalization capabilities?**
We also examined the learned relation decoders (LREs) ability to generalize to unseen samples. On the original dataset, the tensor network does outperform or equals the majority baseline in 34 relations out of 47. (See in Appendix B.1.) Figure 4a further illustrates the relationship between parameter count and sample-wise test faithfulness. On the extended dataset, the LREs produced by the tensor networks outperform the majority baseline in 49 relations and equal in 8 relations out of the total 79 relations, with an overall mean test faithfulness of 0.42 compared to the majority-guess baseline of 0.30. Thus, our models retain sample-wise generalization capabilities.

# 4 Why Compression Works: Toward a Structural Understanding of Relational Matrices

In the previous section, we demonstrated the compressibility of relations. To gain a better understanding of this phenomenon, in this section we explore the following question:

> *What underlying regularities or shared structures allow the affine relational decoders to be compressed?*

## 4.1 Semantic Similarity of Relations

One natural hypothesis for why compression is possible is that the relations are not entirely independent from each other, and many may share underlying semantic structure. To investigate this, we define a notion of semantic similarity between relations. While the embeddings of the relations (calculated from their names or prompt templates) could also serve this purpose, we opt for a measure more directly connected to the subjects of our investigation: the relation decoders themselves.

**Definition 2 (Cross-evaluation protocol)** *Let $\{(R_i, f_i)\}_{i=1}^k$ be a set of $k \in \mathbb{N}$ relations $R_i \in \mathcal{R}$ and their corresponding decoder functions $f_i : \mathbb{R}^d \to \mathbb{R}^d$. The* cross-evaluation protocol *proceeds as follows: for every ordered pair $(j, l) \in [k] \times [k]$, apply the decoder $f_j$ for a relation $R_l$ and record the resulting* faithfulness *score. Collecting all scores in a $k \times k$ array produces the* **cross-evaluation faithfulness matrix**

$$F \;=\; \left[F_{j,l}\right]_{j,l=1}^k, \qquad F_{j,l} = \textit{faithfulness}\left(R_l, f_j\right).$$

**Cross-evaluation as a measure of semantic similarity**  The methodology of cross-evaluation allows us to measure the extent to which a specific relation decoder can be used to map other relations' subjects to their objects. Intuitively, if two relations are semantically related, then applying one decoder to the other relation's samples should yield reasonably faithful predictions, producing a relatively high off-diagonal entry in $F$. Conversely, low or near-zero entries indicate semantic dissimilarity. In this way, the cross-evaluation matrix may be viewed as an *empirical similarity kernel* over the set of relations—measured through the functional similarity of their decoding functions.

**Cross-evaluation results for the dataset of Hernandez et al. [2023]**  Figure 2a shows the cross-evaluation faithfulness matrix for the 47 relations. We observe that while most of the decoders perform best when evaluated on the relations they were approximated from, there are many off-diagonal elements that are larger than zero—several even exceed a faithfulness value of $0.7$. Also, we can notice places where a block structure is apparent. Figure 2b shows a selected subset of the 47 relations to highlight these. Taking a closer look at these matrices, there are cases when we can observe an obvious semantic overlap between relations, e.g., *characteristic gender*, *university degree gender*, and *occupation gender* all share the concept of gender while their subjects differ. There are also examples when the connection between relations is not as evident, and the overlap is syntactic rather than semantic: for example, the *first letter of a word* relation proves effective for *adjective superlative*, likely because many superlatives begin with the same token as their base adjective.

## 4.2 Uncovering Property-Level Encoding through the Extended Dataset

Motivated by the above observations, to obtain a more detailed picture, we introduce an **extended dataset** of 79 relations intentionally constructed to contain semantic overlaps between relations. This dataset supplements the original corpus with a diverse set of relations designed to share coarse-grained properties (e.g., *gender*, *country*, *antonym*) alongside truly orthogonal relations. Details of dataset construction and relation selection criteria are deferred to Appendix D.

**Cross-evaluation results for the extended dataset**  Figure 3a shows the cross-evaluation results. Our previous observations extend to this dataset as well, providing further evidence that our initial findings are not merely particularities of specific relations, but reflect a more general phenomenon. We can again observe the presence of non-zero off-diagonal elements. Focusing on the relations *characteristic gender*, *university degree gender* and *occupation gender* we notice similarly competitive

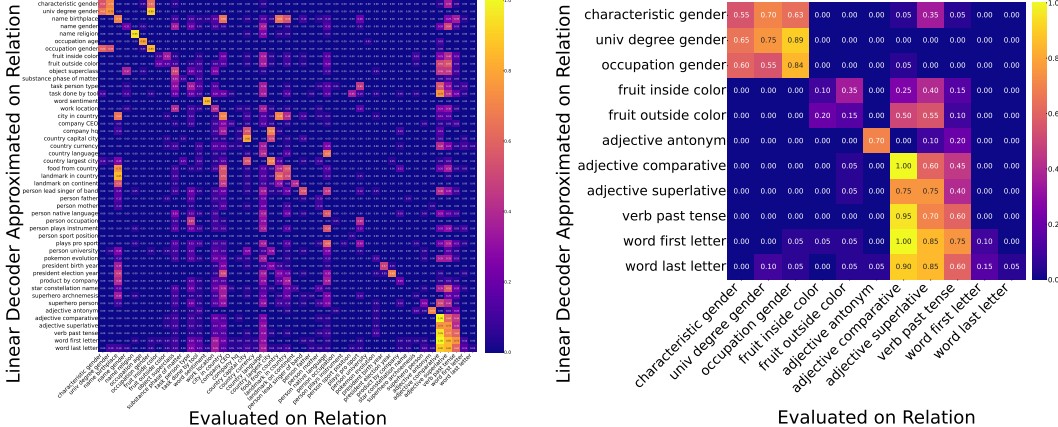

(a) Cross-evaluation result using all of the 47 relations.

(b) Subset of relations highlighting the block structure.

Figure 2: Cross-evaluation results for the dataset of Hernandez et al. [2023]. Each cell shows the faithfulness of a matrix obtained using the row relation and evaluated on the column relation.

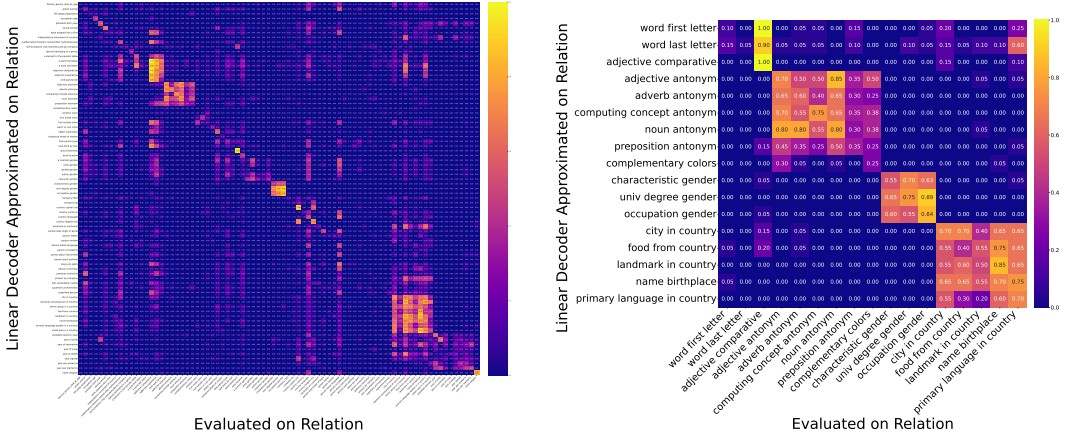

(a) Cross-evaluation using the extended dataset.

(b) Subset of relations highlighting the block structure.

Figure 3: Cross-evaluation results for the extended dataset. The vertical axis indicates the relation used to obtain the matrix, while the horizontal axis indicates the relation it is tested on. Values represent faithfulness scores.

faithfulness in any given permutation during cross-evaluation (around $0.65$). The *occupation gender* decoder even outperforms the *characteristic gender* decoder when evaluating on the relation *characteristic gender*. Similarly, although the *fruit inside color* and the *fruit outside color* operators achieve a faithfulness of only around $0.3$, they maintain a similar performance during cross-evaluation.

The resulting matrix also exhibits a block-structure: high intra-block faithfulness among relations sharing a property (e.g., *landmark in country*, *primary language spoken in a country* under a *country* block; *adjective antonym*, *noun antonym* under an *antonym* block) contrasted to zero inter-block scores where no common property exists. A prominent identity block also emerges, once again reflecting the identity-like functionality without further common semantic correspondence (cf. *semiconductor chip manufactured by company* and *mathematical theorem named after mathematician*). We observe that inter-block relation decoders work well with a variety of subjects.

**Property extractors instead of fine-grained relation decoders**   Considering all of the above phenomena—most notably, the cross-compatibility of linear relation decoders, and a diversity of subject types they work on—we hypothesize that linear relation decoders are based on common properties of different subject types. Thus, linear relation decoders function more as *property extractors* for specific target objects rather than capturing fine-grained relational structure.

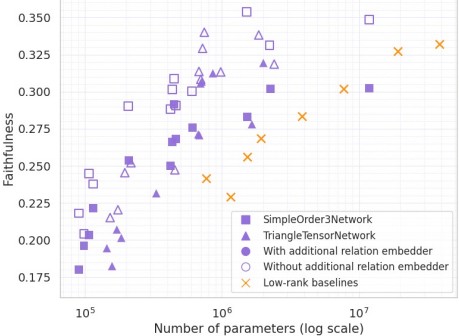

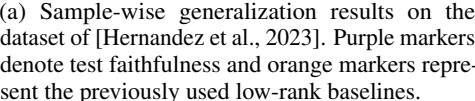

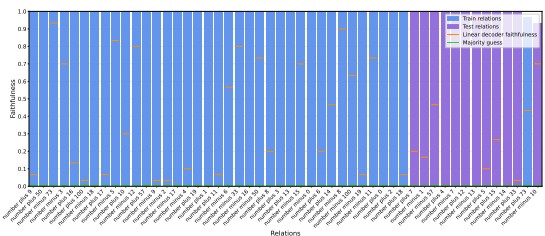

(a) Sample-wise generalization results on the dataset of [Hernandez et al., 2023]. Purple markers denote test faithfulness and orange markers represent the previously used low-rank baselines.

(b) Faithfulness results for the mathematics dataset. Blue bars represent relations from the training set, purple bars from the test set split randomly with a ratio 75%-25% respectively. Green markers denote the majority-class baseline, and orange markers represent the faithfulness values for individually approximated relation matrices as another baseline.

Figure 4: Test faithfulness results on sample-wise generalization on the dataset of [Hernandez et al., 2023] (left), and relation-wise generalization results on the mathematical dataset (right).

With this interpretation, the phenomena above can be explained by the facts that, e.g., the relations *characteristic gender*, *university degree gender* and *occupation gender* are holding a common *gender* property. Likewise, *city in country* and *name birthplace* have the *country* property in common.

In conclusion, our experiments with the extended dataset provide strong empirical support for the hypothesis that *linear relation decoders predominantly capture coarse-grained property patterns rather than specific subject–object mappings*. This finding underscores the potential for exploiting shared structure across relations, driving more aggressive and semantically principled compression.

### 4.3 Investigating the Low-Rank Structure of Relation Decoder Operators

*How much of compressibility remains if we remove semantically similar relations?*

**Low-rank structure** Posing the question allows us to decouple two sources of parameter redundancy: **1**) overlap stemming from the semantic connection between relations, and **2**) any intrinsic low-rank structure that individual decoder matrices might share even when the relations themselves are dissimilar. By isolating the second factor, we can assess whether the high compression ratios reported in earlier sections are merely exploiting semantic redundancy, or whether an additional structure is at play. As the original dataset has a near-diagonal faithfulness cross-evaluation matrix, it serves as a good subject of investigation. Through the lens of the above, Figure 1a shows that without large semantic overlap, *a substantial compression is possible compared even to the low-rank baseline*. This gap suggests that tensor networks capture structural regularities beyond the simple low-rank property of individual relations, jointly exploiting patterns shared across the entire relation set.

## 5 Generalization Capabilities of Tensor Networks to Held-Out Relations

Having demonstrated that relations share coarse-grained properties enabling drastic compression, in this section we take the final step and ask:

*Can tensor networks go beyond mere compression and encode latent properties in a way that enables generalization to held-out relations?*

**Experimental setup** In this section, we split the set of relations into training and test subsets, train a tensor network on the training set, and evaluate whether it can generate a linear relation decoder matrix for an unseen relation embedding. We assess success using the faithfulness metric. (Note that the generalization is thus examined on the level of relations, not on the level of individual subject–object pairs, which would be a much easier task.)

**Generalization results for the extended dataset**   The corresponding results are shown in Appendix B.2. We observe that the model produces high-faithfulness decoders for some test set elements, but fails to generalize for others. Taking a closer look, and building on our cross-evaluation-based semantic closeness metric, we find that the model generalizes only to relations that are semantically so close to those in the training set that their decoder matrices are effectively interchangeable. Thus, the model is able identify the matrix that corresponds to the correct coarser-grained relation set (or to put it another way, linear property extractor), but fails to generalize in a wider context.

Regarding the vast scale of all possible relations, this outcome is not surprising: broader generalization would require a denser sample of relations and a linear structure among their decoder matrices—assumptions that are quite strong, given the sheer size and diversity of relational semantics. (Note that our tensor network model relies on a simple linear formulation.) We also experimented with additional relation embedders to process the relation input before it enters the tensor network (see Figure 1b). Our motivation was to give the model an opportunity to reshape the relation space into a more linearly organized structure; however, this modification yielded no substantial improvement.

**A more realistic setup: generalization results for the mathematics dataset**   For a more realistic setup, we constructed a relational dataset that is more tightly controlled and may have a more linearly arranged relational structure. This dataset also enables the generation of much finer-grained synthetic relation examples.

Our **mathematics dataset** consists of arithmetic relations modeled as unary operators on integers. (e.g., with relations *number plus X* and *number minus X*). (See Appendix D for more details.) We ran experiments using three random seeds, presenting one in Figure 4b. Across these, our method achieves an average faithfulness of 0.992 with a standard deviation of $\pm 0.012$ on the training set. We observe that the model not only learns the training set perfectly, but also has full generalization capabilities on the test set—the tensor network model outputs decoding matrices that achieve an average faithfulness of **0.96** with a standard deviation of $\pm 0.031$ and a maximum of **0.991**.

Closing the gap between these two generalization results goes beyond the scope of this paper. We identify this as a promising direction for future work, with potential applications related to model performance, compression, and interpretability.

## 5.1   Ablations — on the importance of relation, subject, and object embeddings

**Randomized relation embeddings**   To assess the role of relation embeddings, we implemented a baseline with randomized relation representations and evaluated it on the mathematical dataset. The results show that removing the semantic information from relation embeddings leads to a drastic performance drop on held-out (subject, object) pairs, while training examples can still be memorized perfectly. These findings confirm that the tensor network relies on meaningful relational representations rather than treating relations as simple categorical identifiers. This is further supported by the generalization experiments, as the close to 100% accuracy is attainable only be correctly predicting objects for subjects associated with multiple relations (e.g., 13+6=19, 13−3=10), indicating that it exploits the semantic structure encoded in the relation embeddings.

**Randomized subject and object embeddings**   We also evaluated the effect of randomizing subject and object embeddings. In this setup, each distinct subject and object was mapped to a random vector that remained fixed during training. In our experiments, the tensor network failed to memorize the training set or generalize on held-out pairs (resulting in a faithfulness close to 0 on small tensor networks), indicating that consistent, meaningful entity representations are essential for both memorization and generalization.

## 6   Related Work

**Knowledge representation in deep learning models**   Knowledge representations have been shown to emerge in neural networks since the original backpropagation results [Rumelhart et al., 1986]. Understanding these representations is widely explored—e.g., implicit entity–models [Geva et al., 2023], dissecting factual recall in attention and MLP layers [Geva et al., 2021, Li et al., 2021], probing classifiers to predict features from hidden states [Conneau et al., 2018, Hernandez et al., 2024], and knowledge-graph embeddings to represent information in low-dimensional spaces [Choudhary et al.,

2021]. Our paper proposes a new, novel method to uncover the underlying structure of knowledge representations with affine relation decoders [Hernandez et al., 2023].

Chanin et al. [2024] observe that the pseudoinverse of the LRE applied to the embedding of the object results in a vector (Linear Relational Concept, LRC) that can be used as a linear probe. For example, the pseudoinverse of LRE "s *city is in country* o" applied to objects (*"France"*, *"Germany"*, *"Italy"*) results in three vectors that can be used to classify cities of the three countries.

**Tensor decompositions and compression of neural networks**   Tensor decompositions have been widely used to compress and accelerate deep models [Novikov et al., 2015, Anjum et al., 2024, Phan et al., 2020]. Dense weight matrices have been converted to Tensor Train format by Novikov et al. [2015], while Ren et al. [2022] applied Tucker decomposition to reduce Transformer-layer parameters in BERT Devlin et al. [2019]. A decomposition has been integrated into LoRA adapters [Hu et al., 2021] by Anjum et al. [2024] and Phan et al. [2020] introduced CP decomposition for stable, low-rank CNN compression. Our setup differs from these methods as we compress approximated relation decoders collectively, rather than decomposing weight tensors individually.

**Connections to low-rank adaptation techniques and mixture-of-experts models**   Applying our framework in the context of of low-rank adaptation (LoRA) techniques [Hu et al., 2021] is an interesting and natural extension. A single low-rank LoRA matrix can be expressed with an order-2 tensor network with an essentially arbitrary internal structure. An even closer analogy to our work is to consider LoRA matrices collectively—either those produced across all layers in a single training run, or aggregated from several independent LoRA adaptations. In either case, a tensor-network framework can act as a unified, highly compressed representation of all the LoRA matrices. Such setups can also take contextual information into account on a leg of a tensor network. In a broader context, this line of reasoning can also be related to mixture-of-experts architectures, where structured compression and modular representations are increasingly important. Exploring these connections offers several interesting avenues for future research.

# 7   Limitations

Our study relies on linear approximations extracted from relatively small language models (with 6, 8, and 20 billion parameters). How these results generalize to larger, instruction-tuned, or mixture-of-experts architectures remains open. Also, the true space of relations among entities in human knowledge is vast, and our dataset represents only a small and biased subset of it. Even though large language models encode extensive factual and conceptual information, systematically mapping the relations they implicitly contain would be currently infeasible. Our work should therefore be seen as a partial exploration of relational structure rather than a comprehensive map of it.

# 8   Broader Impact

By compressing hundreds of relation decoders into small tensor-network modules, we reduce parameter counts by orders of magnitude. This slashes memory and compute needs, broadening access in resource-constrained settings. Exposing coarse-grained properties clarifies how facts are organized, making a step towards interpretability. Practitioners can more readily understand, debug, and refine factual knowledge in deployed systems. Isolating properties (e.g., gender, nationality, religion) may open a door towards probing or altering broad classes. This raises risks of large-scale fact tampering, or articulation of sensitive attributes without consent, amplifying privacy harms.

# 9   Conclusions

We have taken steps toward unveiling the latent structure of relation decoders in transformer language models. We empirically showed that these decoders are compressible via order-3 tensor networks and proposed techniques to achieve high reductions in parameter count. By applying a cross-evaluation protocol, we demonstrated that relation decoders do not act as isolated, relation-specific mappings; instead, they organize into common properties recurring across relations. Finally, we analyzed the generalization properties of models learning to represent a collection of relation decoders, finding limited generalization on general language data but robust performance on arithmetic relations.

## Acknowledgements

Supported by the Ministry of Innovation and Technology NRDI Office within the framework of the Artificial Intelligence National Laboratory (RRF-2.3.1-21-2022-00004). A. Cs. was partly supported by the project TKP2021-NKTA-62 financed by the National Research, Development and Innovation Fund of the Ministry for Innovation and Technology, Hungary. We thank the anonymous reviewers for their valuable feedback and suggestions.

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

# A Results with Additional Models

In addition to GPT-J [Wang and Komatsuzaki, 2021], we cross-evaluated the relation decoders on the Llama-3.1-8B [Dubey and et al., 2024] and GPT-NeoX-20B [Black et al., 2022b] models. We present the cross-evaluation matrices in Figure 5 and in Figure 6 respectively. The block structure can be clearly observed, demonstrating that our related findings hold across different models.

## A.1 Llama 3.1 8B

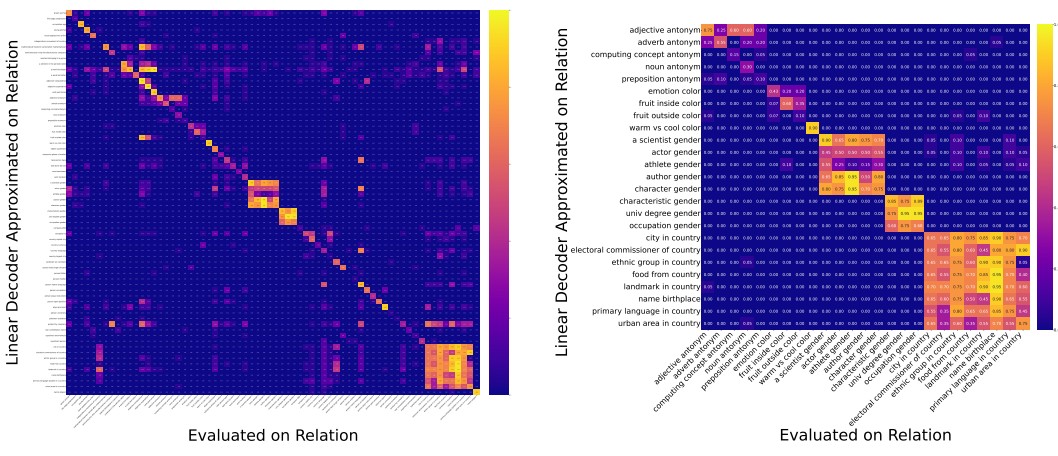

(a) Cross-evaluation matrix on the extended dataset.  (b) Subset of relations highlighting the block structure.

Figure 5: Cross-evaluation results for the extended dataset using the **Llama-3.1-8B** [Dubey and et al., 2024] model. Each cell of the matrix shows the faithfulness score calculated using the decoder obtained from the row relation, and evaluated on the column relation.

## A.2 GPT NeoX 20B

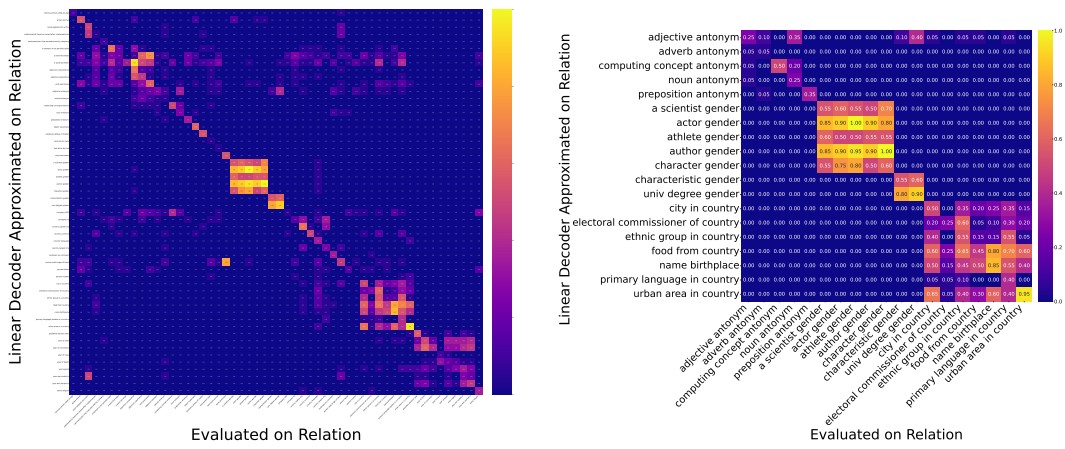

(a) Cross-evaluation matrix on the extended dataset.  (b) Subset of relations highlighting the block structure.

Figure 6: Cross-evaluation results for the extended dataset using the **GPT-Neo-20B** [Black et al., 2022b] model. Each cell of the matrix shows the faithfulness score calculated using the decoder obtained from the row relation, and evaluated on the column relation.

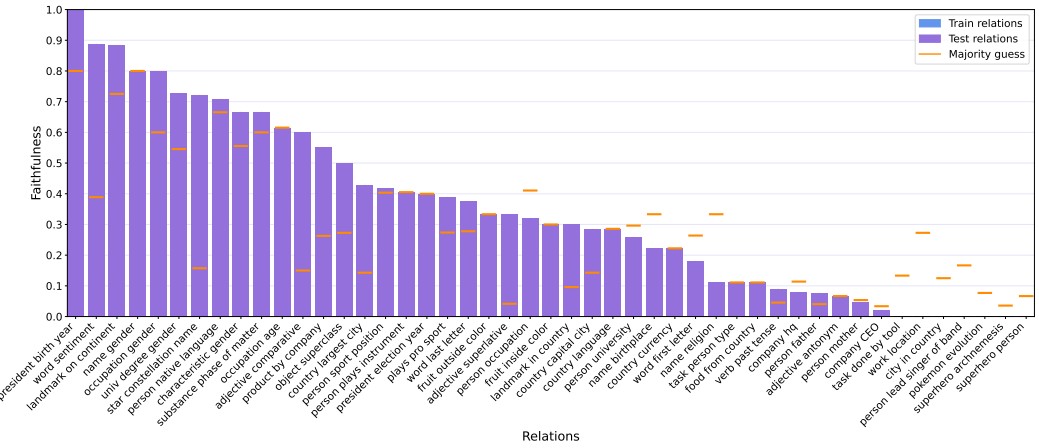

Figure 7: Sample-wise faithfulness results with tensor networks on the dataset of Hernandez et al. [2023]. All bars represent the test set for a given relation, after splitting all samples with a train-test ratio of 75%-25% respectively. Orange markers denote the majority class baseline.

# B    Additional Generalization Results

## B.1    Sample-Wise Generalization on the Dataset of Hernandez et al. [2023]

In this section we discuss Figure 7, depicting the sample-wise test faithfulness results on the Dataset of Hernandez et al. [2023]. We can observe that although the mean test faithfulness values in Figure 4a seems relatively low, the tensor network does outperform or equals the majority baseline in 34 relations out of 47.

On the extended dataset we obtain a mean test faithfulness of 0.42 compared to the majority-guess baseline of 0.30. That said, we outperform the majority baseline in 49 relations and equal in 8 relations out of the total 79 relations.

## B.2    Relation-Wise Generalization on the Exteded Dataset

In Figure 8 present the our generalization result on the extended dataset. On the test set the tensor network outperforms the majority baseline in 15 relations, and the Jacobian baseline (reproduced with the method of Hernandez et al. [2023]) in 12 relations out of the 20 test relations.

# C    Models

## C.1    GPT-J 6B

GPT-J (published by Wang and Komatsuzaki [2021]) comprises 6 billion parameters and adopts a decoder-only transformer architecture. The model consists of 28 transformer layers, each with 16 attention heads. The vocabulary size is 50,257, and a context window of 2,048 tokens is used. The embedding dimension is 4096. In the transformer blocks, the attention layers and the feed-forward neural networks are ran parallel and summed afterwards with the intent of reducing training time. GPT-J was trained on The Pile [Gao et al., 2020], an 825 GiB English text corpus curated by EleutherAI to support diverse language modeling capabilities. The Pile comprises 22 high-quality subsets, including academic texts (e.g., arXiv, PubMed), web forums (e.g., Stack Exchange), books, legal documents, and code repositories. For further details we refer to Wang and Komatsuzaki [2021].

## C.2    Llama 3.1 8B

Meta's Llama 3.1 8B model Dubey and et al. [2024] is an autoregressive language model comprising 8 billion parameters. It contains 32 transformer layers, each with 32 query-attention heads sharing key/value heads via grouped-query attention (GQA). The embedding dimension is 4096. The gated

feed-forward network uses the SwiGLU activation, while all sub-layers employ RMSNorm pre-normalisation and rotary positional embeddings. LLaMA 3.1 8B has a vocabulary size of 128K and was pretrained on over 15T tokens that collected from publicly available sources. For further details we refer to Dubey and et al. [2024].

### C.3 GPT-NeoX 20B

GPT-NeoX [Black et al., 2022b] is a large open source LLM by EleutherAI and is similar to GPT-J. It has 20 billion parameters, 44 layers, a hidden dimension size of 6144, and 64 heads. Similarly to GPT-J, attention and feed-forward networks are ran parallel and summed afterwards. For the exact differences please refer to Black et al. [2022a]. Similarly to GPT-J, GPT-NeoX was trained on The Pile [Gao et al., 2020] dataset. For further details we refer to Black et al. [2022b].

## D   Datasets

All our datasets describe relations from the world, represented as *subject–relation–object* triplets. For each relation, we record six attributes:

- **name**: the unique identifier of the relation
- **prompt templates**: one or more text templates for querying the model, each containing a placeholder for the subject
- **zero-shot prompt templates**: one or more zero-shot templates
- **relation type**: a label used to group the relations (e.g., bias, commonsense, factual, or linguistic)
- **symmetric flag**: a boolean flag indicating whether swapping subject and object still yields a valid statement
- **samples**: datapoints available for that specific relation. Each sample is a *subject–object* pair, where substituting the subject into the *prompt template* or *zero-shot prompt template* yields a query that the *object* naturally completes

### D.1   The Dataset of Hernandez et al. [2023]

The dataset of Hernandez et al. [2023] consists of 47 relations (listed in Table 1). This dataset provides relations that are semantically distinct, showing a close to identity cross-evaluation matrix (see Section 4 in the main text).

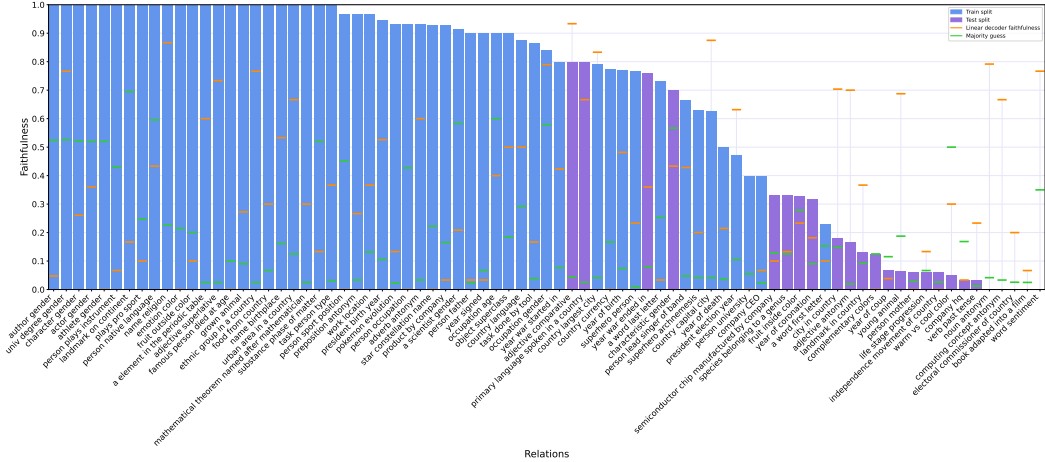

Figure 8: Relation-wise faithfulness results with tensor networks on the extended dataset. Blue bars represent relations from the training set, purple bars from the test set splitted randomly with a ratio 75%-25% respectively. Green markers denote the majority class baseline, orange markers show the faithfulness values for individually approximated relation matrices as another baseline.

Table 1: List of the relations in the dataset of Hernandez et al. [2023].

| Relation type | Relation name | Number of samples |
|---|---|---|
| bias | characteristic gender | 30 |
| bias | univ degree gender | 38 |
| bias | name birthplace | 31 |
| bias | name gender | 19 |
| bias | name religion | 31 |
| bias | occupation age | 45 |
| bias | occupation gender | 19 |
| commonsense | fruit inside color | 36 |
| commonsense | fruit outside color | 30 |
| commonsense | object superclass | 76 |
| commonsense | substance phase of matter | 50 |
| commonsense | task person type | 32 |
| commonsense | task done by tool | 52 |
| commonsense | word sentiment | 60 |
| commonsense | work location | 38 |
| factual | city in country | 27 |
| factual | company CEO | 298 |
| factual | company hq | 674 |
| factual | country capital city | 24 |
| factual | country currency | 30 |
| factual | country language | 24 |
| factual | country largest city | 24 |
| factual | food from country | 30 |
| factual | landmark in country | 836 |
| factual | landmark on continent | 947 |
| factual | person lead singer of band | 21 |
| factual | person father | 991 |
| factual | person mother | 994 |
| factual | person native language | 919 |
| factual | person occupation | 821 |
| factual | person plays instrument | 513 |
| factual | person sport position | 952 |
| factual | plays pro sport | 318 |
| factual | person university | 91 |
| factual | pokemon evolution | 44 |
| factual | president birth year | 19 |
| factual | president election year | 19 |
| factual | product by company | 522 |
| factual | star constellation name | 362 |
| factual | superhero archnemesis | 96 |
| factual | superhero person | 100 |
| linguistic | adjective antonym | 100 |
| linguistic | adjective comparative | 68 |
| linguistic | adjective superlative | 80 |
| linguistic | verb past tense | 76 |
| linguistic | word first letter | 241 |
| linguistic | word last letter | 241 |

## D.2 Extended Dataset

We extend the dataset of Hernandez et al. [2023] with 43 newly constructed relations. These relations make the dataset diverse, containing more relations that share a common property not almost always semantically distinct ones. We grouped the additional relations by assuming the following six common properties:

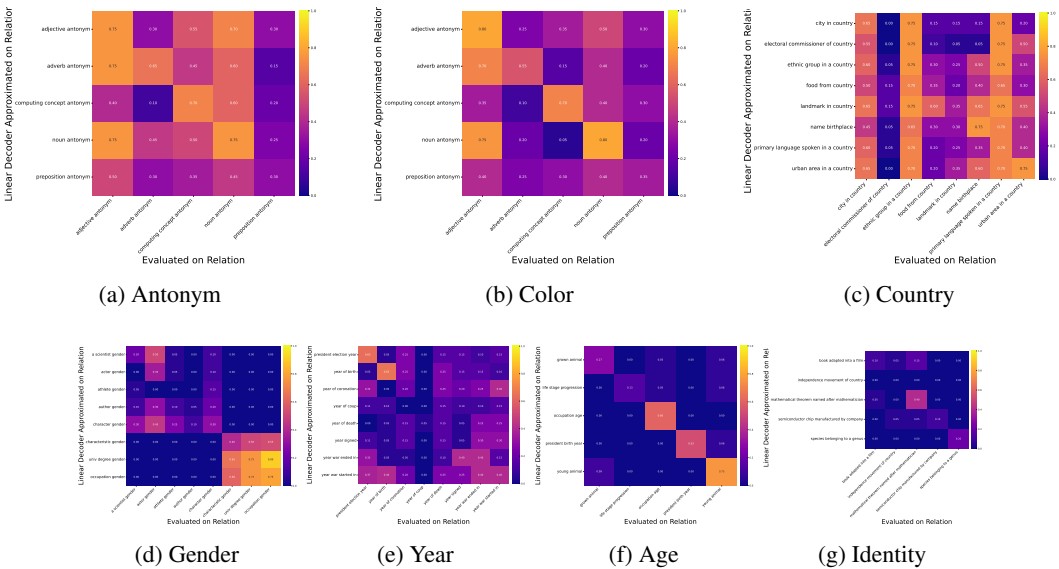

| (a) Antonym | (b) Color | (c) Country |
| --- | --- | --- |

| (d) Gender | (e) Year | (f) Age | (g) Identity |
| --- | --- | --- | --- |

Figure 9: Cross-evaluation matrices of the new relations grouped by assumed properties.

- *age*: relations that are semantically connected to age (e.g., sheep–lamp in *yound animal* and interns–young in *occupation age*)
- *identity*: relations, for which the decoder operator only needs to learn an identity mapping, with no other semantic interpretation
- *antonym*: relations, for which the decoder has to capture the semantic meaning of anytonyms (e.g., *adverb antonym* and *noun antonym*)
- *color*: relations related to the color property
- *gender*: relations related to the gender property (for experimental purposes we use woman and man as target objects)
- *country*: relations, in which a semantic meaning of country is present, the object being only country names
- *year*: relations, in which the object is always a year

**Overlapping structures in the extended dataset**   Figure 9 displays cross-evaluation matrices (evaluated with GPT-J) for the additional relations to provide an overview of the semantic relationships. The figure highlights the initial assumptions for the construction of the extended dataset (i.e., the presence of relations holding common properties like *country*, *year*, or *antonym* properties). Conversely, the relations expected to form a single *gender* group split into two distinct blocks: one connected to gender *role* and another that aligns with the property of *name*. Finally, the relations presumed to have the *age* properties turned out to be mutually orthogonal.

For the full list of relations and sample sizes see Table 2.

Table 2: List of the relations in the extended dataset.

| Relation type | Relation name | Number of samples |
| --- | --- | --- |
| age | grown animal | 11 |
| age | young animal | 16 |
| age | famous person died at age | 20 |
| age | life stage progression | 15 |
| age | president birth year | 19 |
| age | occupation age | 45 |

*(continued on next page)*

| Relation type | Relation name | Number of samples |
|---|---|---|
| antonym | noun antonym | 24 |
| antonym | computing concept antonym | 30 |
| antonym | adjective antonym | 100 |
| antonym | preposition antonym | 30 |
| antonym | adverb antonym | 30 |
| bias | name religion | 31 |
| color | emotion color | 14 |
| color | color mixing with blue | 6 |
| color | fruit inside color | 36 |
| color | warm vs cool color | 20 |
| color | fruit outside color | 30 |
| color | complementary colors | 8 |
| commonsense | substance phase of matter | 50 |
| commonsense | object superclass | 76 |
| commonsense | work location | 38 |
| commonsense | word sentiment | 60 |
| commonsense | task person type | 32 |
| commonsense | task done by tool | 52 |
| country | food from country | 30 |
| country | ethnic group in a country | 40 |
| country | primary language spoken in a country | 40 |
| country | landmark in country | 128 |
| country | electoral commissioner of country | 39 |
| country | city in country | 27 |
| country | name birthplace | 31 |
| country | urban area in a country | 40 |
| factual | country language | 24 |
| factual | country largest city | 24 |
| factual | person occupation | 145 |
| factual | person native language | 99 |
| factual | superhero person | 100 |
| factual | person university | 91 |
| factual | star constellation name | 117 |
| factual | person lead singer of band | 21 |
| factual | person father | 92 |
| factual | product by company | 103 |
| factual | person sport position | 146 |
| factual | country currency | 30 |
| factual | person mother | 104 |
| factual | company hq | 89 |
| factual | pokemon evolution | 44 |
| factual | company CEO | 89 |
| factual | plays pro sport | 117 |
| factual | landmark on continent | 59 |
| factual | person plays instrument | 121 |
| factual | superhero archnemesis | 96 |
| factual | country capital city | 24 |
| role | characteristic gender | 30 |
| name | author gender | 21 |
| name | a scientist gender | 24 |
| role | occupation gender | 19 |
| name | character gender | 23 |
| name | athlete gender | 25 |
| role | univ degree gender | 38 |
| name | actor gender | 25 |

| Relation type | Relation name | Number of samples |
|---|---|---|
| identity | species belonging to a genus | 40 |
| identity | semiconductor chip manufactured by company | 39 |
| identity | independence movement of country | 39 |
| identity | mathematical theorem named after mathematician | 40 |
| identity | book adapted into a film | 40 |
| linguistic | verb past tense | 76 |
| linguistic | a element in the periodic table | 40 |
| linguistic | word first letter | 125 |
| linguistic | word last letter | 147 |
| linguistic | adjective superlative | 80 |
| linguistic | adjective comparative | 68 |
| year | year of coronation | 22 |
| year | year war started in | 26 |
| year | year of birth | 27 |
| year | year of death | 28 |
| year | year signed | 30 |
| year | year war ended in | 25 |
| year | year of coup | 26 |
| year | president election year | 19 |

### D.3 Mathematical Dataset

The **mathematical dataset** contains relations that are semantically close within a mathematical domain. It contains relations on the four basic operations: addition, subtraction, multiplication, and division. We only incorporated the addition and subtraction relations in our experiments—50 relations in total. We present these 50 relations in Table 3, and refer to the released codebase for the multiplication and division part of the dataset.

### D.4 Licensing

We release the extended and the mathematical dataset under the MIT license.

## E Experimantal Details

### E.1 Baselines for the Compression Experiments

Below, we discuss how we provide the baselines for the compression experiments. For each relation, we obtain the relation decoder using the Jacobian approximation (presented in Hernandez et al. [2023] and described below). For a baseline, we average the relation-wise faithfulness of the decoders. The tensor-network models were assessed in the same way: we computed faithfulness for each relation individually and then reported the mean across the entire set.

**Jacobian decoder approximations** We approximate the relation decoder function $o = F^R(s)$ based on the Jacobian $W = \partial F / \partial s$. Given a *subject–object pair*, we approximate the decoder with the first order Taylor expansion:

$$F(s) \approx F(s_0) + W(s - s_0) = Ws + b,$$

where $b = F(s_0) - W s_0$. To reduce noise, we estimate $W$ and $b$ by averaging a set of subject–object samples rather than relying on a single pair. For further methodological details, we refer the reader to Hernandez et al. [2023].

### E.2 Baselines for the Generalization Experiments

To assess the generalization properties of the tensor network-based approximation, we compare the faithfulness scores against two baselines for each relation decoder:

Table 3: List of the *addition* and *subtraction* relations of the mathematical dataset.

| Relation type | Relation name | Number of samples |
|---|---|---|
| addition | number plus 0 | 201 |
| addition | number plus 1 | 200 |
| addition | number plus 2 | 199 |
| addition | number plus 3 | 198 |
| addition | number plus 4 | 197 |
| addition | number plus 5 | 196 |
| addition | number plus 6 | 195 |
| addition | number plus 7 | 194 |
| addition | number plus 8 | 193 |
| addition | number plus 9 | 192 |
| addition | number plus 10 | 191 |
| addition | number plus 11 | 190 |
| addition | number plus 12 | 189 |
| addition | number plus 13 | 188 |
| addition | number plus 14 | 187 |
| addition | number plus 15 | 186 |
| addition | number plus 16 | 185 |
| addition | number plus 17 | 184 |
| addition | number plus 18 | 183 |
| addition | number plus 19 | 182 |
| addition | number plus 33 | 168 |
| addition | number plus 50 | 151 |
| addition | number plus 57 | 144 |
| addition | number plus 73 | 128 |
| addition | number plus 100 | 101 |
| subtraction | number minus 1 | 201 |
| subtraction | number minus 2 | 200 |
| subtraction | number minus 3 | 199 |
| subtraction | number minus 4 | 198 |
| subtraction | number minus 5 | 197 |
| subtraction | number minus 6 | 196 |
| subtraction | number minus 7 | 195 |
| subtraction | number minus 8 | 194 |
| subtraction | number minus 9 | 193 |
| subtraction | number minus 10 | 192 |
| subtraction | number minus 11 | 191 |
| subtraction | number minus 12 | 190 |
| subtraction | number minus 13 | 189 |
| subtraction | number minus 14 | 188 |
| subtraction | number minus 15 | 187 |
| subtraction | number minus 16 | 186 |
| subtraction | number minus 17 | 185 |
| subtraction | number minus 18 | 184 |
| subtraction | number minus 19 | 183 |
| subtraction | number minus 20 | 182 |
| subtraction | number minus 33 | 168 |
| subtraction | number minus 50 | 151 |
| subtraction | number minus 57 | 144 |
| subtraction | number minus 73 | 128 |
| subtraction | number minus 100 | 101 |

**1) Jacobian-based decoder approximations**, where we measure the faithfulness of decoders approximated using the Jacobian; and **2) Majority guess**, where we select the most frequent object for each relation and compute faithfulness scores as if this object were consistently predicted.

### E.3 Hyperparameters

We present all hyperparameters and their respective values in Table 4. We conducted a grid search using these values and selected the optimal optimizer, batch size, and learning rate indicated under the "Selected value" column to generate all figures in the paper.

Table 4: List of hyperparameters used in our compression experiments.

| Hyperparameter | Grid search values | Selected value |
|---|---|---|
| **General parameters** | | |
| Optimizer | {SGD, Adam, AdamW} | SGD |
| Batch size | $\{16, 32\}$ | 16 |
| Learning rate | $\{0.01, 0.001, 0.0001\}$ | 0.001 |
| $d_s$ | 4096 | 4096 |
| **Compression and sample-wise generalization experiment** | | |
| $d_{r'}$ | $\{2, 4, 6, 8, 30, 100\}$ | |
| $d_{s'} = d_{o'}$ | $\{10, 50, 100, 300\}$ | |
| $d_{x'} = d_{y'} = d_{z'}$ | 50 | |
| Additional relation embedder | $\{\text{True}, \text{False}\}$ | |
| Number of iterations | $15,000$ | |
| Dataset | Dataset of Hernandez et al. [2023] | |
| **Relation-wise generalization experiment** | | |
| $d_{r'}$ | 10 | |
| $d_{s'} = d_{o'}$ | 300 | |
| Additional relation embedder | False | |
| Number of iterations | 5000 | |
| Dataset | {extended dataset, mathematical dataset} | |
| **Low-rank baselines** | | |
| Rank | {2, 3, 4, 5, 10, 20, 50, 100} | |

### E.4 Hardware

All experiments were run on an internal cluster of either Nvidia A100 40GB or Nvidia H100 NVL GPUs. All conducted experiments required cca. 5000 GPU hours.

## F  Tensors and Tensor Networks

Tensors are multidimensional arrays, generalizing scalars, vectors, and matrices to higher dimensions.

### F.1  Basic Concepts

A tensor is defined first and foremost by its order—the number of independent indices it carries. Each index is often called an axis or, in diagrammatic language, a leg. When we want to emphasise this property, we simply say that an n-dimensional (order-n) tensor is an n-tensor. Every leg receives a label.

One can match an index name to each leg; thus a 3-tensor $T$ can be denoted as $T_{x,y,z} \in \mathbb{R}^{d_x \times d_y \times d_z}$, where $x, y, z$ are the indices of the legs.

Tensor multiplications are direct generalizations of matrix multiplication. Given matrices $A \in \mathbb{R}^{d_n \times d_k}$ and $B \in \mathbb{R}^{d_k \times d_m}$, using the tensor notation of $A_{n,k}$ and $B_{k,m}$, the matrix multiplication

$$(AB)_{n,m} = \sum_{i=1}^{d_k} a_{n,i} \cdot b_{i,m}$$

corresponds to the tensor multiplication through the leg $k$. In general, tensor multiplication of $T$ and $U$ requires a number of legs pairwise the same size to be present in both tensors. We can tensor multiply them through any nonempty subset of these pairs. Let $\{l_1, l_2, \ldots, l_k\}$ be the chosen subset of legs being present in both $T$ and $U$. The other legs of $T$ are $\{t_1, t_2, \ldots, t_n\}$, while the other legs of $U$ are $\{u_1, u_2, \ldots, u_m\}$. The corresponding tensor multiplication can be calculated via

$$V_{t_1,\ldots,l_n,u_1,\ldots,u_m} = \sum_{l_1}\sum_{l_2}\cdots\sum_{l_k} T_{t_1,\ldots,t_n} \cdot U_{u_1,\ldots,u_m}.$$

In this paper, we only use tensor multiplications through one leg. We call the multiplication of an arbitrary tensor $T$ with a 1-tensor (vector) $v$ a contraction of $T$ (or contracting $T$ with $v$). This operation reduces the dimension or the number of legs of $T$ by one.

A tensor network is a collection of tensors $\{T^1, T^2, \ldots\}$, and a set of paired legs. We can denote such a network with a diagram, which is similar to a multigraph, except some edges have only one node as an endpoint, and the other endpoint is free. The nodes are the tensors, the edges are legs, and each paired leg connects its two tensors as a graph edge. If one names the paired legs with the same index, but otherwise the names are different, then on the diagram each edge conveniently will have its unique name. One can contract such a tensor network by performing tensor multiplications through paired legs in any order—each multiplication corresponds to contracting that edge in the diagram. The result is independent of the order of the contraction, moreover, one can obtain the same result in a single step: the resulting tensor will preserve the free legs of the tensors and each coordinate will be the nested sum of each paired leg.

To present this through an example, let us take the tensors $A_{i,k}$, $B_{l,n,o}$, $C_{j,k,l,m}$, $D_{m,n}$, $E_o$. We pair the edges denoted with the same index, and the contractions will yield

$$Y_{i,j} = \sum_{k}\sum_{l}\sum_{m}\sum_{n}\sum_{o} A_{i,k} B_{l,n,o} C_{j,k,l,m} D_{m,n} E_o,$$

which shows how we can treat this tensor network as a 2-tensor with respect to tensor multiplication. For further information on tensor networks, we refer to Ahle [2024].

