# OpenReview forum: "The Structure of Relation Decoding Linear Operators in Large Language Models"
_NeurIPS.cc/2025/Conference — NeurIPS 2025 spotlight_

### Official Review · Reviewer_svzd · 2025-06-24

**Clarity:** 4
**Significance:** 3
**Originality:** 3
**Rating:** 5
**Confidence:** 3

**Summary:**

The manuscript investigates the collection of Linear Relation Embedding (LRE) matrices within Transformer language models—principally GPT-J-6B.
The authors (i) visualize the latent structure of hundreds of relation-specific decoders through a novel semantic proximity metric, and (ii) show that the entire set of matrices can be compressed to a rank-3 tensor network with negligible loss of faithfulness.
Experiments span three datasets—original (47 relations), expanded (≈100 relations), and a newly constructed mathematical set (hundreds of relations). Despite a reduction of several orders of magnitude in parameter count, the compressed model preserves or even improves faithfulness and acts as a coarse “property extractor.” Generalisation remains narrow: strong transfer occurs mainly for arithmetically well-structured relations, while performance on semantically distant or one-to-many relations is limited.

**Questions:**

- Have you benchmarked rank-4 or TT-SVD variants, and how do they trade off compression versus fidelity?
- Could the compressed tensor serve as a base for LoRA-style fine-tuning to further cut memory?

**Ethical Concerns:**

["NO or VERY MINOR ethics concerns only"]

**Limitations:**

- One-to-many and many-to-many relations remain challenging.
- Scaling to the full diversity of real-world relations is not yet feasible.

**Paper Formatting Concerns:**

No concerns

**Quality:**

3

**Strengths And Weaknesses:**

Strengths
- Treating relation decoders as a family and clustering them by property similarity is an original perspective that sharpens our understanding of relational structure in LMs.
- A rank-3 tensor network attains equal or higher faithfulness than individual matrices while reducing parameters by several orders of magnitude.

Weaknesses
- Limited model diversity – Only a single GPT-J checkpoint is analysed; applicability to other architectures (e.g., Llama-2, Falcon) and languages remains untested.

---

> ### Author Rebuttal · Authors · 2025-07-31
>
> We are grateful for your valuable and supportive feedback. Below, we provide clarifications and address your questions.
>
> **Model diversity:**
>
> While the main text indeed presents experiments using GPT‑J, the supplementary material already includes the cross‑evaluation experiment also with Llama 3.1 8B and GPT‑NeoX 8B, where we observe essentially the same results as for GPT-J.
> For the camera-ready version:
> - We will extend this analysis and include all of the remaining experiments on Llama 3.1 8B and GPT‑NeoX 8B.
> - We will also explicitly reference these additional results in the main text to ensure readers can easily locate and interpret them.
> - We will also include a figure comparing performance across the different language models.
>
> **Language diversity:**
>
> While our method is language-agnostic in principle, evaluating its performance across different languages would require curated cross-lingual relational datasets, which are beyond the scope of this paper. Nonetheless, we recognize that a thorough multilingual analysis is an important and promising direction for future research.
>
> **TT-SVD and order-4 tensor networks:**
>
> Our experimental setup is specific to order-3 tensors (it requires a subject, relation and object embeddings on the free legs). We considered adding additional contextual information via a fourth leg, but this extension is beyond the scope of the present paper.
>
> Thank you for suggesting TT-SVD. We hypothesize that the supervised signal provided by the cross-entropy loss in end-to-end training plays a crucial role in guiding what information is retained and compressed within the tensor. Since our training method already proved effective in practice, we did not pursue alternative training or tensor factorization approaches.
>
> **LoRA:**
>
> Thank you for bringing up the idea of applying our framework in the context of LoRA. We agree this is a compelling and promising direction, for several reasons:
>
> - A single low-rank LoRA matrix can be expressed with an order-2 tensor network with basically an arbitrary internal structure.
>
> - A setup which is more analogous to our paper is to consider LoRA matrices collectively. We might collect the low‑rank matrices produced by a single LoRA run across an entire network. Alternatively, we could take the matrices obtained from several independent LoRA trainings. In either case, a tensor‑network framework can act as a unified, highly compressed representation of all the LoRA matrices. Such setups can also take contextual information into account on a leg of a tensor network.
> - More broadly, we find this line of thinking highly relevant to mixture-of-experts models, where structured compression and modular representation are increasingly important.
>
> We appreciate the suggestion, we will mention this in the paper. We believe it opens up several interesting avenues for future work.
>
>
> We thank you again for the feedback and we hope the clarifications above address all remaining concerns.

---

### Official Review · Reviewer_Ho1f · 2025-07-01

**Clarity:** 3
**Significance:** 3
**Originality:** 3
**Rating:** 4
**Confidence:** 4

**Summary:**

This paper studies the structure of relation embeddings inside LMs. Past work
has found that pre-trained LMs seem to implicitly implement
Hinton-and-Paccanaro-style linear relational embeddings (in the sense that, for
a given relation, you can find a linear map that transforms subject embeddings
into object embeddings). Here, we're interested in understanding the structure
of this embeddings space, in particular whether it's implicitly low dimensional.
The paper explores a couple of different low-dimensional tensor network
parameterizations and finds that all of them significantly improve decoding
accuracy while reducing the number of parameters needed to represent the full
set of relations. Additional experiments partly explain this by showing that
some relations are highly "transferable" (in the sense that applying the
"university degree stereotypical gender" relation to occupations frequently
yields the "occupation stereotypical gender").

I don't have a huge number of comments---overall I thought this was a neat
empirical paper and likely to be of interest to people working on NN
interpretability. That said, there are a couple of technical points that would
strengthen the confidence of my recommendation.

First, what is being optimized in the equation on line 160, and what is being
evaluated in Fig 1a? Is faithfulness being measured on a set of training (input,
output) pairs, or on a set of held-out pairs? Are the target outputs
(1) ground-truth values, (2) obtained from model predictions, (3) or obtained by matching
the predictions of LREs extracted using the gradient-based method of Hernandez
et al? Even if the outputs are different, are the same inputs being used to
train the tensor networks and the linear decoder baseline?
It's quite surprising to me how much faithfulness improves over the
linear decoder baseline in Fig 1a, and I'm trying to understand whether this
results from a difference in training data / objectives or just from the act of
compressing an existing model.

Second, how much should we interpret these results as telling us something about
*language model* representations versus linear embeddings in general? The number
of relations being compressed here is on the order of the number of embeddings
in the representation space, so it doesn't seem implausible that we would
achieve high compressibility even starting from random embeddings or embeddings
from a much weaker model (e.g. word2vec). The thing that was surprising about
the original Hernandez et al. paper was not that you can embed all these
datapoitns, but that an LM's internal computations themselves seem to correspond
to an LRE that you can read off with just a handful of datapoints. One really
useful set of baselines here would be to (1) randomize all the relation
embeddings, and (2) randomize all the subject / object embeddings, and see how
much this affects your ability to memorize the training set (and with (1), to
generalize to new subject/object pairs)

Finally, a tiny terminological comment---this paper uses "rank" to refer to two
toally different things: the usual linear-algebraic notion of "# of linearly
independent columns" (e.g.  "low-rank embedding") and "# of dimensions in a
tensor" (e.g. "rank-3 tensor"). Consider using "order" for the latter to avoid
confusion.

**Questions:**

see above

**Ethical Concerns:**

["NO or VERY MINOR ethics concerns only"]

**Final Justification:**

The authors provided a number of new results during the discussion period that clarify the role of test set generalization. My score is not higher mainly because the results seem much stronger on compression than generalization, but I continue to have a positive view of this paper and would be happy to see it accepted.

**Limitations:**

see above

**Quality:**

3

**Strengths And Weaknesses:**

see above

---

> ### Author Rebuttal · Authors · 2025-07-31
>
> We sincerely appreciate your thoughtful and constructive feedback. We address your questions in detail and provide the requested clarifications below.
>
> **Training details of Figure 1a:**
>
> Throughout our paper we always use the ground-truth values for the target outputs.
>
> In Figure 1a, we evaluated faithfulness on the set of training (subject, object) pairs. Our intention was to use compressibility as a tool to understand and interpret the decoders’ structural and semantic similarities.
>
> **New results on a (subject, object)-wise train/test split**
>
> To avoid any confusions however, during the rebuttal period we conducted experiments where we made a train/test split on the (subject, object) pairs.
>
> To ensure a comprehensive evaluation, we performed experiments on the original dataset as well as the extended and mathematical variants. These results show that on the mathematical dataset the faithfulness value on the test set is close to perfect, 0.97. On the original and extended datasets, we observe weaker generalization: the model exceeds the majority-guess baseline in only about half of the relations and falls short in the remainder, yielding an overall test-faithfulness mean of 0.31. We attribute this drop in performance to having too little training data for our end-to-end training setup (which is naturally expected to be prone to memorization-related issues). This is not the case in the mathematical dataset however: sufficient amount of samples enable this strong generalization (both sample-wise and relation-wise).
>
> In the camera-ready version of our paper we will elaborate on this level of generalization as well, and include a new figure and analysis with the above content.
>
> **Baselines**
>
> In Figure 1a, we use the entire set of (subject-object) pairs for a relation. By contrast, the linear decoder baseline follows the Jacobian‑based procedure of Hernandez et al., which estimates each LRE from a couple of examples (in particular, we used 8 examples). We selected this baseline to provide a more canonical reference point. We side with the reviewer that our evaluation could benefit from other baselines as well. We outline several options below that we will include in the camera ready.
>
> **Additional baseline: separately trained LREs**
>
> Two natural options are:
> - Full-rank LREs: we train the LREs for each given relation separately.
> - Low‑rank LRE variants: we train the LREs for each given relation separately and use a low-rank parametrization.
>
> During the rebuttal period we implemented both of these additional baselines to provide a clearer picture. We trained both of these new baselines in the similar end-to-end manner that we used throughout the paper, and also implemented the (subject, object) train/test splitting method discussed above.
>
> **Tensor networks outperform the “naive approach”**
>
> Unsurprisingly, the results show that indeed separately trained LREs attain high faithfulness—we have an enormous amount of parameters.
>
> Not just more importantly but perhaps even more surprisingly, even with these strong baselines in place, the tensor‑network achieves a better faithfulness‑per‑parameter trade‑off. In particular, the tensor network delivers higher faithfulness than low-rank LREs with even less parameter count.
>
> For example:
> | Model                       | Parameter count | Mean test faithfulness |
> |----------------------------|-------------------|------------------:|
> |  Low‑rank LRE (rank = 5)   |   2,048,500        | 0.84              |
> |  Low‑rank LRE (rank = 10)  |   4,097,000        | 0.86              |
> |  Low‑rank LRE (rank = 20)  |   8,194,000        | 0.86              |
> |  Tensor network (core size: 47x100x100)  |   1,481,912       | 0.97    |
>
> **Additional baseline: randomize relation embeddings**
>
> During the rebuttal period, we have implemented the suggested baseline of randomized relation embeddings and tested it on the mathematical dataset.
> These results indicate that the absence of the semantic information in relation embeddings results in a drastic performance drop on held-out (subject, object) pairs, whereas training samples can be memorized perfectly.
>
> We would like to note that the generalization experiments (Figure 4) already demonstrate that the tensor network do make use of the relation embeddings, otherwise, it could not succeed in predicting the object for subjects that correspond to more than one relations in the math dataset (e.g., 13+6=19, 13-3=10). The fact that it exploits even the semantic structure (and do not simply treat relation embeddings as simple discriminative labels) is demonstrated by the positive generalization results on the relation-wise held-out test set.
>
> **Additional baseline: randomize subject/object embeddings**
>
> We also conducted the suggested experiment with randomized subjects and object embeddings.
>
> Specifically, we mapped each distinct subject and object to a random vector and used this embedding throughout the training. In our experiments, the tensor network failed to memorize the training set and to generalize on the held-out pairs as well (resulting in a faithfulness of 0).
>
> **Changes to the manuscript**
>
> We plan to implement the following changes to the manuscript:
> - We will include the sample-wise train/test split results, with a new figure and brief analysis.
> - We will add clarifications regarding the baselines (as discussed above) in a new subsection of Section 3 and in the technical details of Section 2.
> - We will update Figure 1a to reflect the full-rank and low-rank LRE baselines.
> - We will discuss the training with randomized relation and (subject, object) embeddings in a new subsection of Section 3.
> - Thank you for caching the rank/order terminology issue. Will adopt your suggestion in the revised version.
>
> We thank you again for the constructive feedback, and we hope we have addressed all your concerns. We remain available to discuss any further questions during the discussion period.

---

> > ### Comment · Reviewer_Ho1f · 2025-08-02
> >
> > Thank you for the very detailed response. The fact that the main experiments are just about memorizing the test set makes me a little less excited about the results, but I'm still positive overall about the paper. In any case I'd encourage you to make that fact very clear in both the presentation of the method and the results, and consider using the generalization eval numbers from the rebuttal as the main results in Figs 1 and 4. The other proposed changes all sound good & address the corresponding issues from the original review. It's pretty interesting that some of the non-math relations do exhibit good generalization, and it would be very interesting to know whether those correlate with the original Hernandez et al. cross-relation differences in faithfulness (which is apparently related to training data frequency https://openreview.net/pdf?id=EDoD3DgivF).

---

> > > ### Author Response · Authors · 2025-08-02
> > > **Crucial update on generalization**
> > >
> > > **Updated results with (subject, object) train/test split!**
> > >
> > > We are delighted to share that continued the experiments that we have implemented during the rebuttal period and found that **tensor networks do generalize well even (subject, object)-wise across all three datasets**.
> > >
> > > More precisely, the results show that tensor networks outperform the baselines not just in terms of the tradeoff between faithfulness and parameter count, but in the mathematic dataset even in overall faithfulness as well.
> > >
> > > We hope these results provide clear support for our paper’s assessment and dispel any of your remaining doubts.
> > >
> > > Results on the original dataset:
> > >
> > > | Model                       | Parameter count | Mean test faithfulness |
> > > |----------------------------|-------------------|------------------:|
> > > |  Low‑rank LRE (rank = 5)   |   1,925,590       |   0.81        |
> > > |  Low‑rank LRE (rank = 10)  |     3,851,180  |      0.82        |
> > > |  Low‑rank LRE (rank = 20)  |   7,702,360   |     0.83     |
> > > |  Low‑rank LRE (rank = 50)  | 19,255,900 |     0.83      |
> > > |  Low‑rank LRE (rank = 100)  |  38,511,800 |    0.83       |
> > > |  **Tensor network (core size: 50x6x50)**  |   **449,276**    |  **0.84**    |
> > >
> > > Results on the extended dataset:
> > >
> > > | Model                       | Parameter count | Mean test faithfulness |
> > > |----------------------------|-------------------|------------------:|
> > > |  Low‑rank LRE (rank = 5)   |   3,236,630        | 0.76              |
> > > |  Low‑rank LRE (rank = 10)  |   6,473,260        | 0.77              |
> > > |  Low‑rank LRE (rank = 20)  |   12,946,520        | 0.77              |
> > > |  Low‑rank LRE (rank = 50)  |   32,366,300       | 0.78              |
> > > |  Low‑rank LRE (rank = 100)  |   64,732,600       | 0.79              |
> > > |  **Tensor network (core size: 50x6x50)**  |   **449,276**      |  **0.80**  |
> > >
> > >
> > > Here, we reiterate and extend the results on the mathematical dataset:
> > >
> > > | Model                       | Parameter count | Mean test faithfulness |
> > > |----------------------------|-------------------|------------------:|
> > > |  Low‑rank LRE (rank = 5)   |   2,048,500        | 0.84              |
> > > |  Low‑rank LRE (rank = 10)  |   4,097,000        | 0.86              |
> > > |  Low‑rank LRE (rank = 20)  |   8,194,000        | 0.86              |
> > > |  Low‑rank LRE (rank = 50)  |   20,485,000       | 0.86              |
> > > |  Low‑rank LRE (rank = 100)  |   40,970,000      | 0.86              |
> > > |  **Tensor network (core size: 50x6x50)**  |    **449,276**    |   **0.88**    |
> > > |  **Tensor network (core size: 47x100x100)**  |   **1,481,912**       | **0.95**    |
> > >
> > > Models were trained until convergence.
> > >
> > > We hope that the sample-wise and relation-wise generalization achieved with tensor networks, together with our cross-evaluation experiments, provide a firm foundation for the paper’s acceptance.

---

> > > > ### Comment · Reviewer_Ho1f · 2025-08-05
> > > >
> > > > Thanks! But now I'm a little confused---what changed between these results and the results you reported in the original rebuttal?
> > > >
> > > > > "On the original and extended datasets, we observe weaker generalization: the model exceeds the majority-guess baseline in only about half of the relations and falls short in the remainder, yielding an overall test-faithfulness mean of 0.31. We attribute this drop in performance to having too little training data for our end-to-end training setup (which is naturally expected to be prone to memorization-related issues).

---

> > > > > ### Author Response · Authors · 2025-08-06
> > > > >
> > > > > Originally, we had an explanation to this inconsistency, but investigating this further we identified an oversight in our updated implementation. This implementation issue arose specifically in this experiment and only during the rebuttal period. When you posted your comment, our experiments had just concluded, and in our excitement over the improved results, we responded prematurely. We sincerely apologize for this and kindly ask you to disregard the earlier comment entirely.
> > > > >
> > > > > To clarify our correct results:
> > > > > - On the mathematical dataset test faithfulness consistently reaches over 0.9 on a large variety of tensor network configurations. This is in line with our previously reported numbers in the rebuttal and clearly demonstrates sample-wise generalization capabilities of the tensor networks.
> > > > > - On the extended dataset the initially reported test faithfulness of 0.31 is a valid result, so what you cite above from our rebuttal remains correct.
> > > > >
> > > > > We would like to emphasise that this oversight does not affect the conclusions of our paper or the conclusions presented in our rebuttal. We genuinely hope that this hurry in implementation during the short one-week rebuttal period does not undermine the rigorous efforts we have devoted to this project over the past year. We greatly appreciate your previous positivity and constructive feedback regardless of this experiment, and hope that you will maintain your favorable view toward our submission.

---

> > > > > > ### Comment · Reviewer_Ho1f · 2025-08-07
> > > > > >
> > > > > > No problem at all---thanks for clarifying! I continue to have a positive assessment of this paper. I do still think it would be good to include the generalization numbers; to put the 0.31 in context, what accuracy would you get from a majority baseline guess on all classes?

---

> > > > > > > ### Author Response · Authors · 2025-08-08
> > > > > > >
> > > > > > > We are happy to hear this, thank you!
> > > > > > >
> > > > > > > As an update on the faithfulness results for the extended dataset: when shuffling the samples before splitting—which is the natural methodology here—we obtain a mean test faithfulness of 0.42 compared to the majority-guess baseline of 0.30. That said, we outperform the majority baseline in 49 relations and equal in 8 relations out of the total 79 relations. We will include this result in the paper instead of the previously reported 0.31, which was trained without shuffling. For clarity, regarding the mathematical dataset, all of the previously discussed conclusions hold.
> > > > > > >
> > > > > > > We will, of course, include and detail this methodology and its results in the paper.

---

### Official Review · Reviewer_Jyaq · 2025-07-03

**Clarity:** 2
**Significance:** 3
**Originality:** 2
**Rating:** 5
**Confidence:** 3

**Summary:**

This paper presents an investigation into the decoding of relational knowledge from LLMs. Building based on the Linear Relational Embedding, this work proposes a rank-3 tensor network model to compress an entire collection of linear decoder functions into a single, compact model. The results show that such compression can largely reduce the model sizes.

**Questions:**

Please see above.

**Ethical Concerns:**

["NO or VERY MINOR ethics concerns only"]

**Final Justification:**

The rebuttal has addressed some of my comments.

**Limitations:**

Yes.

**Quality:**

3

**Strengths And Weaknesses:**

Strengths:
- The paper is quite well written.
- The designed approach is not too complicated.
- The motivation is sound.
- There are some in-depth analysis about why such compression should work.
- There are some interesting, unobvious findings.
- Evaluation has included an extended dataset with more relational types and a mathematical dataset.

Weaknesses:
- Evaluation has mainly focused on GPT-J. Is there any particular reason why this model is chosen?
- Other models, such as Llama, is shown in the supplementary. How generalizable the results are across different models?
- In the supplementary, it has been noted that the experiments took a total of 5000 GPU hours. How long would it take for training one instance of the tensor network model?
- Some GPU memory analysis would be good too, including the effect of the dimension of the embedding space in the original models.
- Many of the figures are too small to see.
- Note: all left quotes need to be reformatted.
- I wonder what the results would be, if instead of using pretrained LLMs, all experiments are conducted on smaller LMs, so that a training from scratch can be conducted using limited datasets. In this way, we can ensure that the LMs have not been exposed to more knowledge present in the test datasets. So we can evaluated whether the compression is not because of it's compressing out knowledge that is not included in the test dataset.

---

> ### Author Rebuttal · Authors · 2025-07-31
>
> We thank you for the positive and valuable feedback; below we address each question and detail the updates in the paper we will make based on them.
>
> - We ran most of our experiments on GPT-J simply because it is the model we started to work with.
>
> - Our cross-evaluation experiments with Llama 3.1 8B and GPT-NeoX 8B exhibit strikingly similar behavior to GPT-J, leading us to anticipate similar outcomes across other experiments as well. In the camera-ready version, we will include these figures also in the appendix, and will refer to these additional results more explicitly in the main text of the paper.
>
> - Regarding the legibility of our figures, we will increase the font sizes and refine labels—we can take advantage of the extra page in the camera‑ready version to do so.
>
> - We will also reformat the left quotes.
>
> - We appreciate the suggestion to train smaller LMs from scratch to eliminate any potential pre‑training exposure to the test data. While this is indeed valuable, the scope of our paper is to decode and compactly represent the knowledge already present in pretrained LMs. Exploring this question would address a promising, but rather different research direction: how relational knowledge is acquired under constrained training.
>
> - Wall-clock time: the models train for about 3 hours on a single H100 GPU. Much of the reported 5000 GPU hours were spent on grid searches and frequent evaluations conducted to support a detailed analysis of model behavior.
>
> - GPU memory: for example, in the case of the simple tensor network, the total parameter count is $d \cdot d_s + d \cdot d_o + d \cdot d_r + d_s \cdot d_r \cdot d_o$, where $d$ is the embedding dimension of the LLM, and $d_r, d_s, d_o$ are the inner dimensions order-3 tensor in the core. Notably, when substantial compression was achieved, the core tensor parameter count $d_s \cdot d_r \cdot d_o$ was significantly lower than that of the projection matrices ($d \cdot d_s + d \cdot d_o + d \cdot d_r$). We will include a brief and exact analysis of the parameter counts and memory usage in the main text, as it would indeed enhance the quality and clarity of the paper.
>
> We thank you again for the constructive feedback, and we hope we have addressed all your concerns. We remain available to discuss any further questions during the discussion period.

---

> > ### Comment · Reviewer_Jyaq · 2025-08-05
> >
> > The rebuttal has addressed most of my comments.

---

### Official Review · Reviewer_NLdX · 2025-07-03

**Clarity:** 3
**Significance:** 3
**Originality:** 4
**Rating:** 5
**Confidence:** 4

**Summary:**

This paper builds on prior work showcasing that the computation involved in relations like "The capital of France is___" (relation=capital, subject=france, object=paris) is well approximated by linear (or affine) maps that generalize across instances of that relation (e.g., the same map works on capital of spain). This work shows that relation decoders can be compressed across different relations that share similar properties, thus viewing these decoders as property extractors rather than single relation decoders. The authors present a novel architecture for training a parameter efficient version of these extractors with tensor networks. The experiments show promising results, but are a little thin in areas. The findings are novel and interesting to the interpretability community.

**Questions:**

N/A

**Ethical Concerns:**

["NO or VERY MINOR ethics concerns only"]

**Final Justification:**

See review. I feel positively about this paper

**Quality:**

3

**Strengths And Weaknesses:**

# Strengths
* The Hernandez et al., 2023 work is very interesting, and related to a body of interpretability work on related structures (both preceding Hernandez, and currently growing). The property centric view on these relations that allows the compression of what were previously collections of individual relation decoders proposed by this paper would be of great interest to this research community.
* Shared representational structure within models is an important finding in any domain. This paper provides a nice example with relational structures beyond the single relation level

# Weaknesses
* Figures 2 and 3 both being devoted to large (slightly illegible) heatmaps is an unfortunate use of space. I think this could be used for more results or visualization that better show the results. Perhaps Figures 2b and 3b become one figure and the 'a' sections go to the appendix?
* Besides the math dataset, the held out relations don't actually seem to get much benefit from the heldout relations. I count about 8 that look to be substantially far above the majority decision and most underperform the single relation linear decoder. As the authors mention, this result isn't surprising. I think it was worth trying it, but since the result is so strongly expected (this was an extremely hard setup), I feel like it's not the best use of space. The result showing the generalization for math capabilities is very interesting

# Missing citations
* The authors should provide the proper citations when discussing prototype theory (Rosch, 1971), as well as provide a citation for Wittgenstein when discussing these ideas, although they are so widely known.
* The authors should cite [Chanin et al., 2024](https://arxiv.org/abs/2311.08968) which extends LREs to concept embeddings. While not redundant with the current work, I think these findings are pretty relevant to this paper. **Specifically, I view the LRCs as property extractors for single relations**, the current paper makes the property extractor idea more salient and introduces compression across relations. Is this a fair account?

---

> ### Author Rebuttal · Authors · 2025-07-31
>
> Thank you for your valuable suggestions. We will apply the following changes to further improve our paper:
>
> - We will include the missing citations.
>
> - To improve readability in 2b and 3b, we will enlarge fonts and refine labels.
>
> - We will move Figure 4a to the Appendix and only cover its content in the main text. In its place, we will add additional visualizations—e.g., a figure comparing the results of the three evaluated models (GPT‑J, Llama 3.1 8B, and GPT‑NeoX 8B).
>
> - Regarding 2a and 3a, we explored multiple alternatives before settling on the current versions. Our view is that these convey crucial global context: they show that the cross‑evaluation matrices are not diagonal—the surprising observation that motivated our investigation into the semantic structure of relations. Without illustrating this bigger picture, we believe we would lose important context for the reader. With the additional page in the camera‑ready version, we believe that there will be sufficient space to keep this figure in the main text.
>
> - Thank you for drawing our attention to Chanin et al. (2024), as you pointed out, they explore a related but different idea. Their work focuses on (relation, object) pairs with LRCs represented ultimately as vectors, whereas our work focuses on the relations, and the LRE functions themselves. For this reason, we would not interpret LRCs as property extractors in our framework, but there is indeed a meaningful interplay between the notions and approaches presented in the two papers. We will add the reference and a brief paragraph to discuss the relation between the two.
>
> We are again grateful for your positive feedback. We hope that our responses fully address your suggestions. We are happy to elaborate on any remaining issues throughout the discussion period.

---

> > ### Comment · Reviewer_NLdX · 2025-08-04
> >
> > Thank you for the replies. I look forward to the new figures and the comparison to Chanin et al. I will maintain my score

---

### Decision · Program_Chairs · 2025-09-17

**Decision:**

Accept (spotlight)

**Comment:**

This is an interesting and well-executed paper that investigates relational knowledge in LLMs and offers several insightful findings. All reviewers spoke positively about the work, and the author rebuttal adequately addressed most of the concerns.

That said, a few limitations remain:

- The studied problem may have a relatively narrow audience within the research community. In some related problems, relational knowledge alone is not a sufficient representation, which suggests that the scope of the proposed interpretation may be limited. Nevertheless, this line of work could potentially open a broader direction in the future.

- The analysis primarily focuses on GPT-J. Extending the study to more modern models would strengthen the generality and impact of the findings.